# Breeding Biofortified Pearl Millet Varieties and Hybrids to Enhance Millet Markets for Human Nutrition

**Mahalingam Govindaraj ***, **Kedar Nath Rai, Binu Cherian, Wolfgang Helmut Pfeiffer, Anand Kanatti** **and Harshad Shivade**

International Crops Research Institute for the Semi-Arid Tropics (ICRISAT), Patancheru 502324, Telangana, India; k.rai@cgiar.org (K.R.); b.cherian@cgiar.org (B.C.); w.pfeiffer@cgiar.org (W.P.); k.anand@cgiar.org (A.K.); HShivade@cgiar.org (H.S.)
* Correspondence: m.govindaraj@cgiar.org; Tel.: +91-40-307-13617

**Abstract:** Pearl millet is an important food crop in the arid and semi-arid tropical regions of Africa and Asia. Iron and zinc deficiencies are widespread and serious public health problems worldwide, including in India and Africa. Biofortification is a cost-effective and sustainable agricultural strategy to address this problem. The aim of this review is to provide the current biofortification breeding status and future directions of the pearl millet for growing nutrition markets. Research on the pearl millet has shown that a large genetic variability (30–140 mg kg$^{-1}$ Fe and 20–90 mg kg$^{-1}$ Zn) available in this crop can be effectively utilized to develop high-yielding cultivars with high iron and zinc densities. Open-pollinated varieties (Dhanashakti) and hybrids (ICMH 1202, ICMH 1203 and ICMH 1301) of pearl millet with a high grain yield and high levels of iron (70–75 mg kg$^{-1}$) and zinc (35–40 mg kg$^{-1}$) densities have been developed and released first in India. Currently, India is growing > 70,000 ha of biofortified pearl millet, and furthermore more pipeline cultivars are under various stages of testing at the national (India) and international (west Africa) trials for a possible release. Until today, no special markets existed to promote biofortified varieties and hybrids as no incentive price to products existed to address food and nutritional insecurity simultaneously. The market demand is likely to increase only after an investment in crop breeding and the integration into the public distribution system, nutritional intervention schemes, private seed and food companies with strong mainstreaming nutritional policies. The following sections describe various aspects of breeding and market opportunity for addressing micronutrient malnutrition.

**Keywords:** biofortification; pearl millet; malnutrition; iron; zinc; market

## 1. Introduction

The inadequate intake of energy-providing organic macronutrients (largely carbohydrate, followed by protein and fat, in that order), leads to under-nutrition, with a consequent feeling of hunger. The result of chronic hunger, reflected in such apparent physical manifestations as being underweight, wasting and stunting, has long been debated as a food security issue in various global fora [1]. Unlike the macronutrients mentioned above, which are consumed in larger quantities for proper growth and development, there are several micronutrients which are needed in trace amounts, but they play vital roles in various physiological functions. The deficiencies of these micronutrients do not lead to obvious hunger effects; they lead to all adverse physical manifestations, such as those arising from the deficiencies of macronutrients. Thus, these micronutrient deficiencies are termed as hidden hunger. Deficiencies of iron (Fe) and zinc (Zn) have been reported to be the most widespread, affecting more than two billion people worldwide, mostly in the developing countries [1,2].

Micronutrient malnutrition, primarily the result of a poor quality of diets or a poor intake of vitamins and minerals, affects more than 2 billion people in developing countries, especially women and preschool children [2,3]. The costs of these micronutrient deficiencies in terms of lives lost, the adverse effect on the economic growth and the poor quality of life are very huge and more staggering in developing countries, including India. Therefore, considerable global development efforts are underway to improve the health of poor people by breeding staple food crops enriched with essential micronutrients, so called biofortification. This is a multidisciplinary, sustainable and cost-effective approach to bring the full potential of crop improvement and nutrition science to bear on the persistent problem of micronutrient malnutrition. Pearl millet is highly nutritious, and largely grown under rainfed conditions in India (8 m ha) and Africa (18 m ha) [4]. It supplies 80–90% of the calories for several millions of poor people in the world [5]. In India, being the highest per capita consumption by a rural population, especially in the western Rajasthan and Gujarat, this contributes more than 50% of the cereal consumption in these regions [6], and it is also consumed in other parts of Gujarat, Rajasthan, Maharashtra and Haryana [7].

HarvestPlus is a CGIAR Challenge Program, leading a global effort to breed and disseminate micronutrient-rich crop varieties inducing pearl millet through a partnership in developing countries. While the development of OPVs (100%) continues to be the thrust research area in Africa, the development of hybrids (70%) is the primary focus in India [4]. Pearl millet as such is a high-iron crop with a fairly high Zn content, higher than rice and wheat; however, not all available cultivars have a high Fe and Zn content. So far, the crop breeding programs in ICRISAT and the National Agricultural Research System (NARS) in India and Africa have largely been focused on the grain yield and components traits (including the biotic stress tolerance), and less emphasis was given to nutritional quality traits (iron and zinc) in the core line/cultivar development process. Consequently, there is a narrow range of such micronutrients exhibited in most of the released cultivars (46–56 mg kg$^{-1}$ Fe) that are being used for consumption in poor households [8]. For instance, all the commercial hybrids had 42 mg kg$^{-1}$ mean iron and 31 mg kg$^{-1}$ mean zinc content [8]. Therefore, the thrust of the pearl millet biofortification research targeted for India is on the development of high-yielding and high-Fe hybrids, since the entire efforts in the public and private sector are geared toward hybrid development [4]. This effort has been underway in the ICRISAT biofortification program by utilizing the existing high-Fe hybrid parents and advanced breeding lines identified from the mainstream breeding program to serve the immediate objective of developing high-yielding and high-Fe hybrids. Biofortification research has gradually shifted from the detection of variability through breeding high-Fe cultivars using existing high-Fe lines to the testing and delivery of biofortified hybrids realized in India. Meanwhile, in Africa, 100% OPV cultivation and the fast-track development of OPV using high-Fe populations are in progress [4]. All these screening, breeding and market opportunities are briefly discussed in the following sections.

## 2. High-Throughput Micronutrient Phenotyping

The success of a breeding program depends on the precision phenotyping efficiency through high throughput tools. This is a primary need of the biofortification research and is a key to speedy progress in identifying high-Fe/Zn lines from large set of germplasm and gene pools. Atomic Absorption Spectrophotometry (AAS) and Inductively Coupled Plasma Optical Emission Spectrometry (ICP-OES) techniques (both are destructive methods) are highly used by researchers, and their results are reproducible for grain Fe and Zn densities. However, breeding for micronutrient dense cultivars needs the screening of a large number of genetic material, such as germplasm collections, elite lines, segregating populations, hybrids etc.; and phenotyping for micronutrients though destructive techniques involves a high analytical cost and breeding resources. The recent study using an advanced technique X-ray Fluorescence Spectrometry (XRF) estimates of Fe and Zn densities from diverse pearl millet materials showed a highly significant and positive correlation with ICP values (r = up to 0.97; $p < 0.01$ for Fe and r = up to 0.98; $p < 0.01$ for Zn) [9]. Hence, high-throughput, non-destructive and low-cost quantitative

technique XRF are now being used in several laboratories for the initial screening of large genetic material. The final validation of those top-ranking entries based on the XRF value can be further re-evaluated using ICP to obtain precision estimates for wider reporting and documentation [10].

## 3. Genetic Variability for Micronutrients

Genetic variability is very critical asset to breeders to start with any trait specific breeding. With the high throughput screening facilities, several thousands of pearl millet samples were analyzed for grain Fe and Zn density. This is largely contributed to scale the available genetic variability for these two micronutrients in working germplasm and breeding materials at ICRISAT. Several studies at ICRISAT showed a wide range of variability for grain Fe and Zn densities in diverse breeding materials such as *Iniari* germplasm accessions (51–121 mg kg$^{-1}$ Fe; 46–87 mg kg$^{-1}$ Zn), population progenies (18.0–135.0 mg kg$^{-1}$ Fe; 22.0–92.0 mg kg$^{-1}$ Zn), inbred parents (30.3–102.0 mg kg$^{-1}$ Fe; 27.4 mg to 84.0 mg kg$^{-1}$ Zn), hybrids derived from diverse inbreds (25.8–80.0 mg kg$^{-1}$ Fe; 22.0–70 mg kg$^{-1}$ Zn) and commercial hybrids (31.0–61.0 mg kg$^{-1}$ Fe, 32.0–54.0 mg kg$^{-1}$). Most of these genetic materials with high Fe and Zn levels were entirely or largely based on *Iniari* germplasm [11–13]. *Iniari* refers to early-maturing and large-seeded land races found in adjoining parts of Togo, Ghana, Benin and Burkina Faso [14]. A core collection will be evaluated to explore if *non-iniari* sources of high Fe can be identified to diversify the genetics based on high-Fe materials. A recent validation of advanced breeding lines (seed-parents and restorer parents' progenies) showed that about 128 progenies (81 seed-parents and 47 restorer-parents) were highly adapted to the field condition with the Fe density of 64–133 mg kg$^{-1}$ in seed-parents progenies and of 56–139 mg kg$^{-1}$ in restorer-parents progenies. Of these, 53 seed-parent progenies exceeded 85 mg kg$^{-1}$ Fe density and had 46–80 mg kg$^{-1}$ Zn density. Commercially available cultivars (not bred for micronutrients) had a mean of 42 mg kg$^{-1}$ Fe [8]. The variability for grain minerals in this crop opens the opportunity for breeding high iron breeding lines and hybrid parents and thereby high-iron cultivars for improved human nutrition in millet consuming populations.

## 4. Biofortification Breeding Approach

The pearl millet biofortification breeding program at ICRISAT has taken a three-pronged breeding phase-I, II and III. The first phase is a short-term approach dealing with traits genetics, germplasm screening and creating genetic variability. The second phase is the medium-term approach consisting of validating identified high-iron and zinc breeding lines and hybrid parents from the regular breeding program to develop fast-track biofortified variety/hybrids. The third phase consisted of long-term objective- development high-Fe/Zn breeding lines and hybrid parents and its genetic diversification through steady mainstreaming micronutrient traits at ICRISAT and NARS breeding programs. In brief, these micronutrients are largely governed by additive genes; thus, biofortification breeding approaches are the same as for as any other quantitative traits. Previous reports also indicate the greater importance of additive gene action for grain Fe and Zn in pearl millet [12,15,16]. The heritability of these micronutrients was very high and not much influenced by the environments [17–19]. The pedigree breeding method is a most common method in pearl millet breeding which deals entirely with the progenies derived from bi-parental crosses, and this method is well described by Andrews et al. [20]. This method also utilizes composites as a base population that has the potential to accelerate the genetic gains for the yield in hybrids and widen the genetic base of hybrid-parents and cultivars. However, composites will be used for pedigree breeding of hybrid parents only when they fulfil the basic traits required of hybrid parents. Pearl millet has different cytoplasmic sterility systems (CMS); however, A1, A4, and A5 are widely used. The adoption of cytoplasmic genetic male sterility falls under the standard three-line system (A, B, and R) to produce a hybrid seed. Currently, A1 and A4 are exploited under the commercial breeding programs in India. All the pearl millet hybrids so far developed in India are based on A1 CMS, while the biofortification program uses both A1 and A4 CMS to diversify the new cultivar base. Currently, the pearl millet biofortification is in a gradual transition from fast-track breeding (breeding phase II) to genetic diversification and mainstreaming

(breeding phase III). The development of biofortified fast-track hybrids/varieties at ICRISAT is briefly described in Figure 1. Currently ICRISAT demonstrated the combining micronutrients and grain yield by conventional breeding approaches. The fast-track approach explained in Figure 1 uses existing moderate to higher Fe/Zn content breeding lines and a population that is not bred for the micronutrient content. Breeding parental lines (seed- and restorer-parents) bred for high Fe density as a target trait will address the long-term objective of breeding hybrids, despite high-Fe levels. In this direction, 174 high-Fe early-generation progenies (B × B progenies and R × R progenies) have been developed, which in trials conducted at Patancheru had shown > 90 mg kg$^{-1}$ Fe density and 36 to 72 mg kg$^{-1}$ Zn density. Based on a preliminary study, the identified common and overlapping Quantitative Trait Loci (QTL) for Fe and Zn densities are in LG3 (chromosome 3) [21]. Consequently, the substantial variation for Fe and Zn is available in elite breeding populations, and more emphasis is required on developing diagnostic makers for screening segregating materials in the future.

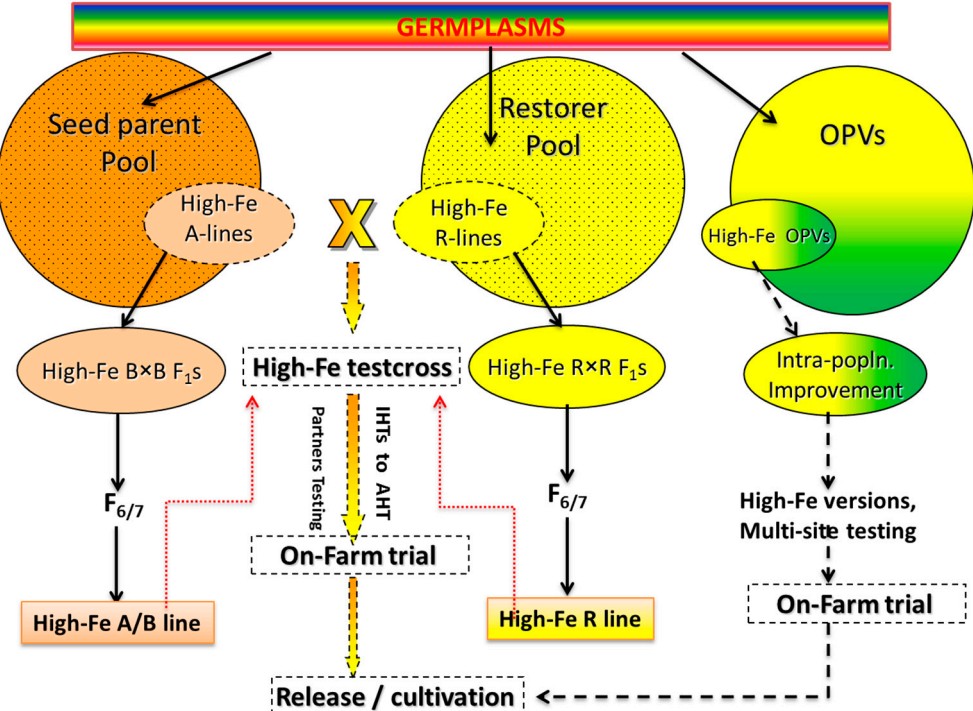

**Figure 1.** Fast-Track breeding approach followed at ICRISAT for biofortified hybrid development in India and OPV development in West Africa.

## 5. Biofortified Cultivar Release and Adoption

In continuation with the germplasm screening, we identified high-Fe progenies, and breeding lines were used to develop initial crosses for building hybrid parents and the OPV development, respectively. In the short and medium-term objectives, identified high-Fe hybrid parents and populations contributed to develop high-Fe cultivars for rapid delivery. A world first high-Fe pearl millet variety 'Dhanashakti' was developed by utilizing the intra-population variability within ICTP 8203, an early-maturing, large-seeded, disease resistant and high-yielding open-pollinated variety that has been under cultivation in India since 1990. Dhanashakti is officially released and notified by Central Variety Releasing Committee in 2014 for the cultivation in all pearl millet growing states of India [22]. Based on national testing trials, Dhanashakti had 71 mg kg$^{-1}$ Fe (9% higher) and 2.2 t ha$^{-1}$ grain yield (11% higher) compared to the original. ICMV 221, another largely grown OPV, is also under improvement for Fe. The improved version (ICMV221 Fe 11-2) had 70 mg kg$^{-1}$ Fe density (11% higher), 58 mg kg$^{-1}$ Zn density (9% higher), and a grain yield of 3.2 t/ha (5% higher) compared to original variety. The first

biofortified variety Dhanashakti reached nearly 90,000 famers in India and the introduction of this biofortified variety in West Africa is in progress [23].

The current breeding efforts in the public and private sector are towards hybrid development and indeed supported by ICRISAT through the development and dissemination of a large number and diverse range of improved breeding lines and hybrid parents through Pearl Millet Hybrid Parent Research Consortia (PMHPRC). By utilizing identified high-Fe hybrid-parents from amongst those initially not bred for high-Fe as a target trait, several high-yielding and high-Fe hybrids have been developed, which are at various stages of testing. One such hybrid has been identified and designated as ICMH 1201, which in 48 field trials during 2011–2013 gave a 75 mg kg$^{-1}$ Fe density (similar to ICTP 8203) but had 3.6 t ha$^{-1}$ grain yields (38% higher than ICTP 8203). ICMH 1201 flowered only 3 days later than ICTP 8203, so it falls in the early-maturity group and fits in the same production system. ICMH 1201 and eight other biofortified hybrids developed at ICRISAT and identified by our partners as promising hybrids have been tested over two years in the All India Coordinated Research Project on Pearl Millet (AICRP-PM) trials [24]. At the same time, the commercial production of ICMH 1201 was undertaken by Shaktivardhak Seed Company since 2014 for marketing under its brand name Shakti-1201 [25]. Shakti-1201 has been adopted by 35,000 farmers, in Maharashtra and Rajasthan [24]. Several hybrids with a high-Fe density comparable to a high-Fe experimental hybrid ICMH 1201, with a higher grain yield, were identified at various stages of testing. The grain yield of these hybrids was not comparable but close (85–90%) to the highest-yielding and high-Fe (56 mg kg$^{-1}$ Fe) commercial hybrid 86M86 (from Pioneer Hi-Bred Pvt. Ltd.), but most of these biofortified hybrids flowered a week earlier than 86M86, hence it was suitable for the medium maturity zone. The first wave of biofortified hybrids are AHB 1200 Fe (ICMH 1202), HHB 299 (ICMH 1203), and Phule Maha Shakti (ICMH 1301) officially released for a national level in collaboration with the agricultural universities of Maharashtra (VNMKV and MPKV) and Haryana (CCSHAU). These biofortified hybrids contain more than 70 mg kg$^{-1}$ Fe and 35 mg kg$^{-1}$ Zn (Figures 2 and 3).

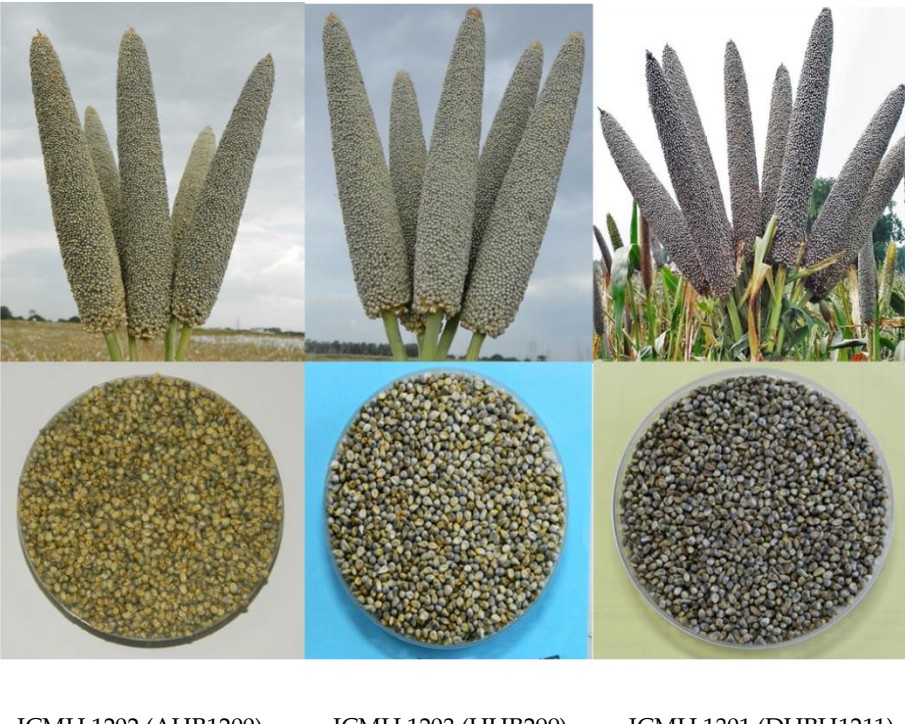

ICMH 1202 (AHB1200)      ICMH 1203 (HHB299)      ICMH 1301 (DHBH1211)

**Figure 2.** Released and notified biofortified high-yielding hybrids in India.

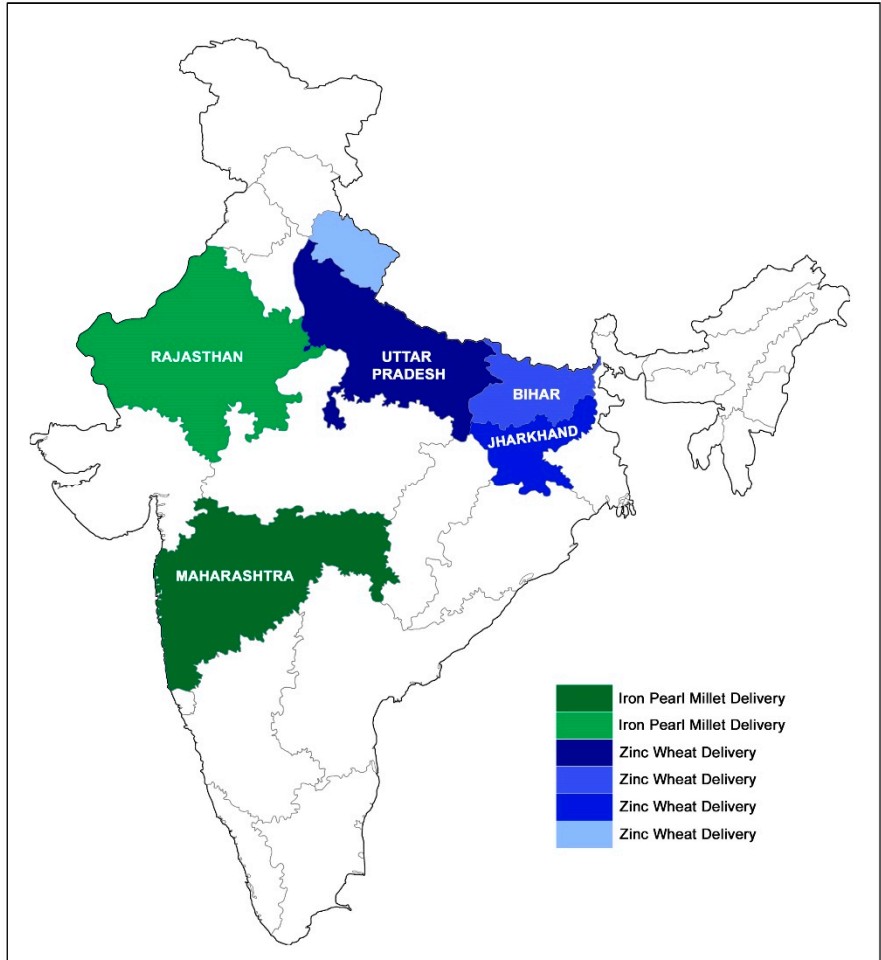

**Figure 3.** Initiation of biofortified pearl millet cultivars cultivation spread in India (Adapted form HarvetPlus-India).

This biofortification product marks the first milestone in the mainstreaming of the grain mineral traits in the cultivar development process and is expected to have a policy in place for the grain mineral bench mark to enable long-term biofortification at all level of hybrid testing and release in India and Africa. With the great support of partners, especially MPKV-Dhule, AICRP-PM, Mondore and Nirmal seeds Pvt Ltd., first high-iron verity Dhanashakti reached regular seed systems that annually send 200–300 kg of breeder seed to national partners; so far they have reached > 60,000 farmers in peninsular India. The truthfully labeled seed (TLS) production of Shakti-1201 is being undertaken by Shaktivardhak Seed Company for commercialization, and it has been adopted by 20,000 ha, mostly in Maharashtra and Rajasthan. Much greater progress in adopting high-Fe hybrids with a high-grain yield is expected in the near future.

## 6. Trade-Off between Yield and Micronutrients

The knowledge of the relationship between micronutrients and yield traits is a prerequisite to combining these two market traits into an improved genetic background. A highly positive and significant correlation between Fe and Zn encourages a concurrent genetic improvement [9,26–28]. Meanwhile, both micronutrients are negatively and non-significantly associated with the grain yield due to the genetic architecture of the breeding lines and cultivar, as they were coming from targeted selections for the yield and not targeted for the micronutrients [16,29]. Interestingly, the biofortified hybrids which are bred for both traits are reflected no in a negative association with the grain yield (Figure 4). The indication of this no relationship between the productivity level of the environments and

the grain yield and micronutrients implies that a higher productivity does not necessarily lower the grain micronutrients. Furthermore, the seed weight is positively correlated with the grain micronutrients, indicating an indirect way for grain nutrients to contribute to the yield potential of a line or hybrid. Therefore, a careful selection should be followed in the initial breeding cycles to achieve a high-yielding and higher micronutrients.

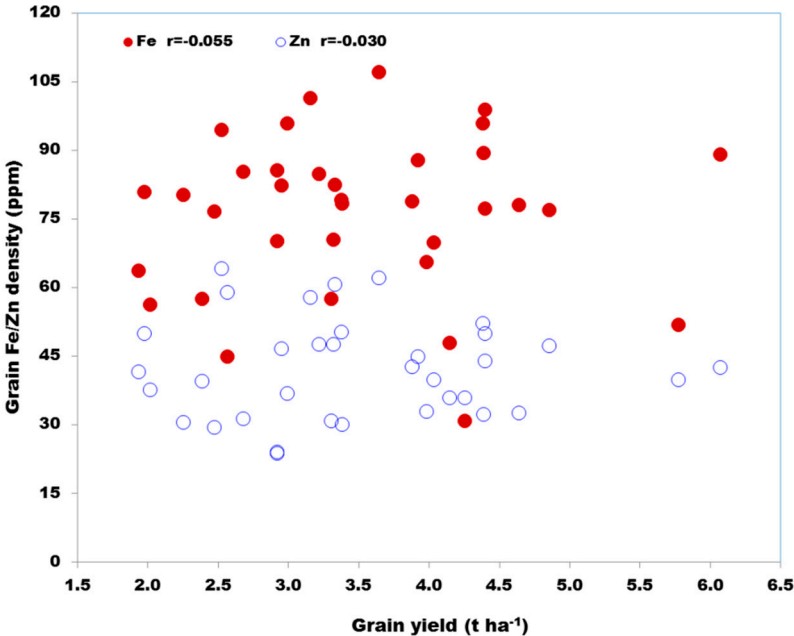

**Figure 4.** Relationship between the Fe/Zn density and grain yield in biofortified hybrid.

## 7. Market Opportunities

In India, strong pearl millet breeding programs exist in both the public and private sectors. The commercial market is largely driven by hybrids (90%) with the private sector playing a key role; hence, biofortification initiatives engage both actors in the mainstreaming iron pearl millet. Private sector seed companies have strong pearl millet research and development programs for hybrids. To ensure long-term sustainability, seed companies must engage in iron pearl millet breeding and establish their own high-iron product lines. Therefore, the HarvestPlus strategy engages seed companies in the genotype-by-environment (G × E) testing of hybrids and inbred lines developed at ICRISAT, and they encourage companies to develop their own high-iron hybrids for commercialization. Over the past 10 years of crop development with NARS and seed companies, many breeding and product testing challenges have been addressed. However, the involvement of seed companies is not up to the target of mainstreaming micronutrients since no policy is in place to support biofortified varieties for public pull and profit marketing (Figure 5). Therefore, revisiting some of the mainstreaming challenges (yellow status) and addressing a lack of policy to pull the nutritious varieties into a marker are a high priority in the future. Private seed companies operate a two-tier distribution system, supplying seed directly to distributors who in turn sell to retailers [30]. Generally, each distributor sells to 40–50 retailers, depending on the location and crops/products. These retailers ultimately sell the seeds to farmers. Pearl millet seeds are sold commercially in 1.5 and 3 kg packs, in attractive primary packaging with mandatory labeling (according to India's Seed Act). The demand for iron pearl millet seed is created by farmer demonstrations, field days, and promotions at points of sales. Nirmal Seed's channel partners and personnel have been trained in nutrition messaging for iron pearl millet, a crucial component in the delivery process [23]. Building product acceptance is further facilitated by the agronomic superiority of recently released high-iron cultivars. Now, biofortification programs are required to collaborate with various actors in food and retail to demand the creation of iron pearl millet grains,

flour, and value-added products. This includes testing specific promotional messages and product benefits, communication channels and their effectiveness, and the selection of the brand name and advertising. With this background, the proposed market channel is given and will be channelized at the country level (Figure 6).

| Task | Key milestone | Status | Timeline | Remarks |
|------|---------------|--------|----------|---------|
| 01 | Partnership establishment | Green | 2011 | Complete strengthening **continued-test** |
| 02 | Initial Product testing | Green | 2012 | Key resource has been **shared** |
| 03 | Advanced product testing | Yellow | 2013 | Project **funding** is not sufficient. |
| 04 | National trial | Yellow | 2014 | Improved **testing and sampling** |
| 05 | Integration and Testing | Green | 2015 | Need **partners involvement** |
| 06 | Initiate Mainstreaming | Yellow | 2016 | Need **implementing strategy** |
| 07 | Hybrid Release | Green | 2017 | Need more **products pipelines** |
| 08 | Release & Accept. Stage | Red | 2018 | Need **policy driven** products |
| 09 | Accept. & consumption | Red | ≥2020 | Need **policy and market** driven products |

**Legend:**
= On schedule/ completed
= problems exist, or re-visiting required
= Schedule has not been, or cannot be sustained (policy)

**Figure 5.** Snap-shot of the pearl millet biofortification program challenges and opportunities in developing countries (India and Africa).

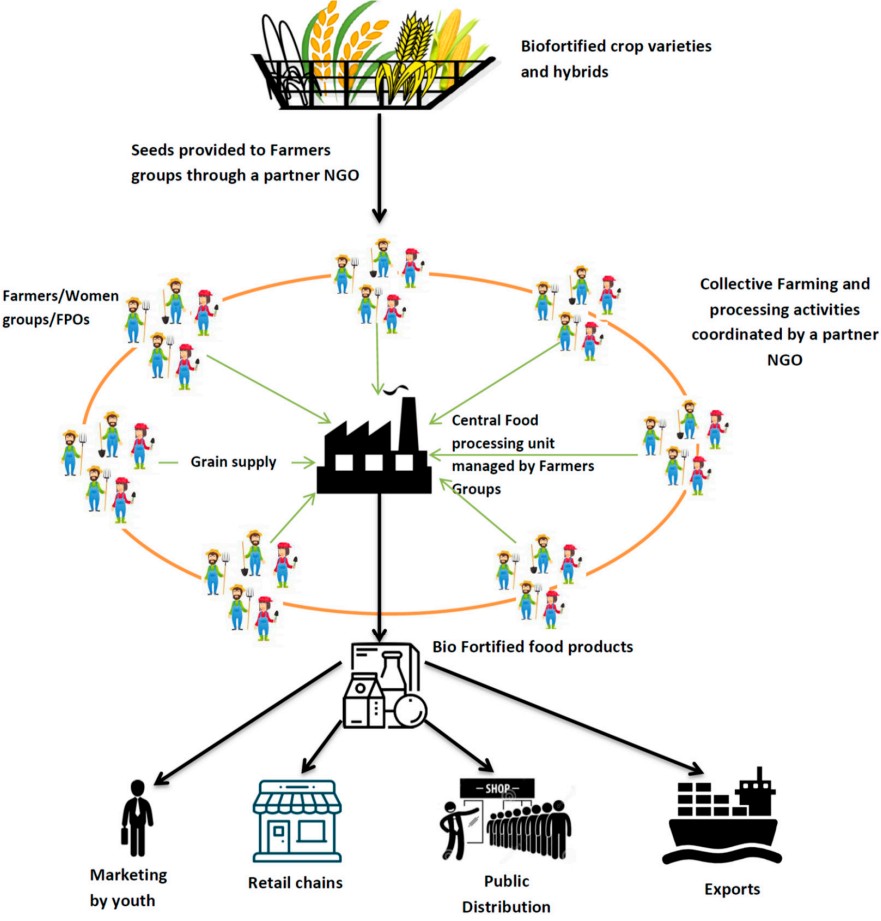

**Figure 6.** Proposed scaling up pathway for biofortified grains from the varieties and hybrid in India (from production to consumption).

### 7.1. Mainstreaming Cultivar Policy

The Indian Council of Agricultural Research (ICAR) has endorsed the inclusion of the minimum levels of iron and zinc that have to be bred into future varieties of pearl millet across the country. The All-India Coordinated Research Project on Pearl Millet decided on a minimum of 42 mg kg$^{-1}$ of iron and zinc 32 mg kg$^{-1}$. Any variety or hybrids to be sold to farmers by public or private seed producers should follow these micronutrient standards, apart from giving a higher yield [31]. The dissemination of these biofortified breeding lines and hybrid parents, and their utilization by user-research organizations (both public and private sector) on a continuing basis, as done so far for the non-biofortified materials, will make biofortified hybrid development a matter of routine and hence significantly contribute to improved nutrition. Mainstreaming should be done.

### 7.2. Market Policy

Unlike other cereals, pearl millet is consumed as a whole grain, and very little processing may be required for new products. Processing technology for pearl millet is yet to be commercialized to improve the edible and nutritional characteristics of millet as well as to improve the grain supply challenges that promote both the urban and the rural household's utilization. Various forms of food products from industries is key to upscale underutilized nutritious crops for better human health. Climate and nutrition smart millet brings the prospects for millet processing entrepreneurs and consumers' willingness irrespective of the purchase power of the rural and urban community.

### 7.3. Smart Food

Pearl millet is a climate smart crop by itself-dryland resilient with high metabolizable energy, high gluten-free protein, and more balanced amino acids. Consequently, this crop has a potential role to play in the smart food initiative taken by ICRISAT (for more detail, visit https://www.icrisat.org/smartfood/), which aims to build food systems where the food is good for you (highly nutritious), good for the planet (climate resilient) and good for the smallholder farmer. In general, several products of millets are available in various countries (http://www.icrisat.org/wp-content/uploads/2018/05/Selected-Millet-and-sorghum-products_1.pdf). The same market linkage can be tapped into for exploring nutri-dense grain and product markets in the future.

## 8. Conclusions and Way Forward

Biofortification is scientifically proven to be a sustainable and cost-effective approach to address malnutrition. This approach targets the root cause of the malnutrition. Combining the yield and micronutrients is highly feasible, like other heritable traits provided the mainstreaming of these traits in the national and international breeding programs. Public and private partners need to take this opportunity to elevate the importance of this nutri-cereal at a national and international level. Efficacy studies provide evidence that the consumption of grains from biofortified varieties would provide bioavailable Fe to meet a fully recommended daily allowance (RDA) in children, adult men and 80% of the RDA in women. In the modern era, genetic biofortification is a way of improving nutrition in staple crops for which enhanced food processing and markets will be the backbone to nutritional security in the future. There is no regulatory mechanism in place to support biofortified crop varieties, and markets are the key for enhancing processed foods, as are related industries. This will also generate market opportunities for farmers. While rice and wheat will continue to play major roles in addressing the problem of food security, biofortified pearl millet has the potential to make significant contributions to the food-cum-nutritional security in dryland poor households. Biofortified cultivars are being developed through conventional breeding, thus grains and foods produced from such cultivars do not face the possible challenge of food regulations and consumer acceptance.

**Author Contributions:** Conceptualization, M.G.; Writing original draft, M.G., A.K., B.C.; Data collection and tables, H.S.; Review and editing, M.G., K.N.R., B.C. and W.H.P.

**Funding:** This research was supported by funding from HarvestPlus Challenge Program of the CGIAR. It was carried as part of the CRP on GLDC and A4NH.

**Acknowledgments:** This work was financially supported by HarvestPlus Challenge Program of the CGIAR.

**Conflicts of Interest:** The authors declare no conflict of interest.

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
