# Peer review of "Breeding Biofortified Pearl Millet Varieties and Hybrids to Enhance Millet Markets for Human Nutrition"

_agriculture, doi:10.3390/agriculture9050106_

Round 1

Reviewer 1 Report

Dear Authors, 

This review article is very interesting as micronutrient deficiency is a major problem in developing countries around the world. It is thoughtful to put together a review to include all the progress made in breeding for biofortified pearl millet. However, there are some suggestions below that can improve the manuscript:

1.     The abstract is too lengthy. Make it precise and include a statement for the motivation of this review. It will be helpful if the authors summarize lines 85 – 89 in the abstract.

2.     Precision phenotyping? Do you mean high-throughput phenotyping (HTP)? If that is the case, please change it to HTP

3.     Is genetic variability determined using HTP? If not, include genetic variability section first and then mention that quantifying micronutrients is a tedious process and describe HTP increases the capacity for evaluating diverse germplasm.

4.     What is the genetic architecture of Zn and Fe concentrations? Are these traits oligogenic, if so, include a section on the genetic architecture of these traits? It will be helpful to mention if there are any major effect QTLs and the heritability of these traits in ICRISAT breeding programs. Based on the genetic architecture, address if marker-assisted backcrossing can be an effective breeding strategy and is it currently utilized in the breeding programs.

5.     In line 134: Additive gene action – is this reported earlier? If so, include references

6.     Started the article focusing on Indian and African pearl millet, in the last few sections the focus was completely shifted to pearl millet production in India. What is the current status of pearl millet hybrid production in Africa? Is Harvest Plus involved in promoting biofortified pearl millet cultivation through NARS in Africa?

Some minor suggestions to improve the manuscript.

Line 75: NARS? Abbreviate NARS

Lines 85 – 89: Too many messages in one sentence. Make this sentence simple

Line 133: Change NRAS to NARS

Author Response

please find attached response to Reviewer#1:

authors

Reviewer 2 Report

Breeding Biofortified Pearl Millet Varieties and Hybrids to Enhance Millet Markets for Human Nutrition

1.       Summary

Outlining the aim of the paper and its main contributions (one short paragraph)

The presented manuscript is a review about biofirtified pear millet and its potential for increasing food security in Asia and Africa. The paper is well organized and the focus is of interest for local and global agriculture. However it requires more detailed information in terms of the agronomical potential of biofortified pear millet and also lacks comparison with other realities around the world.

2.       Broad comments

Highlighting areas of strength and weakness. These comments should be specific enough for authors to be able to respond.

General

The manuscript presented here has its merits in terms of highlighting the agricultural potential of biofortified pear millet and its impact in the food security in specific areas of the globe. However, there are weak points that need to be addressed in all the sections of the manuscript. In general, the manuscript does not present a clear objective and is confuse in the definition of the area of interest, if India, Africa or both. There is also a lack of bibliography review and referencing; many points are stated without a reference, as if it was common sense. The manuscript also misses the comparison of the current advances in the Asia and Africa areas with other parts of the globe. Finally, there is the need of going further in the analysis of agronomical characteristics of the fortified millet.

Abstract

The abstract is too long and the initial paragraphs could be shortened and focus on the main problem to be addressed by the review. Also, there is not a clear objective of the review which could be briefly presented in the body of the abstract. It is important to define the area of interest and how the impact of biofortified millet can differ in Asia and Africa.

Introduction

The Introduction focused on the importance of micronutrients (mainly iron and zinc) in the nutrition of population in India and Africa and how the breeding programs should focus to improve the presence of those micronutrients. It is clear and the reading flows well; however, there are not enough references for many of the affirmative sentences presented. There are only 5 references, mainly focused on the reality of Indian millet production. It should also consider the reality of fortified pear millet in other parts of the globe. Is this biofortified millet already in use in other countries with similar climatic conditions than Asia and Africa?

Precision Phenotyping and Genetic Variability

This section brings more detailed information about the phenotyping and genetic diversity of Fe and Zn contents. There is no mention to the combined analysis of high-Fe and high-Zn cultivars with crop yield and also in terms of the agronomical practices required for biofortified millet. Are there any penalties in yield (tons ha-1) or grain quality (protein and carbohydrate’s amount) when using biofortified pear millet? Are there any requirements of heavier fertilization to boost the natural capacity of the genotypes to deliver high-Fe and high-Zn at the grains?

Biofortification Breeding Approach

This section shows the breeding plan for new lines/cultivars enriched with Fe and Zi. Although there is information about methods, it could be further explained, mainly presenting more details about Figure 1.

Biofortified Cultivar Release and Adoption

This section informs about the current status of breeding for high yield and high-Fe in India and Africa. In my opinion, the information is not clear about the reach of those new released cultivars. Is it spread in India and Africa or only in India. Are there tests for the same cultivars in areas with contrasting climate conditions to prove the advantages are kept in multiple sites?

Market Opportunities

The text is quite superficial in this section and lacks references. For instance, where does the information in Fig. 4 comes from? Is it an observation of the researchers or part of a report from ICRISAT or any other source?

Conclusion and Way Forward

The conclusions are quite general and some sentences a bit vague. I think there is a lack of strong conclusions and/or predicted actions based on recent reports or research papers.

References

There are only 14 references for a 10-page review paper. There is a need of a strong referencing of the information presented in the paper.

With my best regards

Author Response

Dear sir

please find attached response to reviewers #2 .

regards

Authors.

Round 2

Reviewer 2 Report

Revision of version 2

Breeding Biofortified Pearl Millet Varieties and Hybrids to Enhance Millet Markets for Human Nutrition

1.       Summary

Outlining the aim of the paper and its main contributions (one short paragraph)

The presented manuscript is a review about biofortified pear millet and its potential for increasing food security in Asia and Africa. The paper is well organized and the focus is of interest for local and global agriculture. The cover letter and the changes made for the current version have greatly improved the manuscript. Even though, there are still many sentences without reference along the text, what is quite limiting for the quality of a review paper. Please, be careful and make sure there are references for sentences that bring new information to the text. Finally, an English professional review is extremely recommended.

2.       Broad comments

Highlighting areas of strength and weakness. These comments should be specific enough for authors to be able to respond.

General

The manuscript presented here has its merits in terms of highlighting the agricultural potential of biofortified pear millet and its impact in the food security in specific areas of the globe. The current version is better organized than the previous and presents stronger scientific background after the inclusion of more detailed information and further references. The main aim of the review was clarified and better stated. However, I think there is still place for improvement, mainly in the abstract and introduction, as recommended below.

Abstract

The abstract is still too long. It has more than 400 words and sounds more like a short bibliography review. The points raised in the abstract are interesting and come in a good sequence. Although, I think it could be shorten to around 250 words using more direct sentences and linking the ideas better together. It would make the abstract to be easier to read and call the attention of the reader. There are some sentences with lack of sense and grammatical problems and the English must be reviewed.

Introduction

There are still sentences without references in the introduction, mainly at the end of the first paragraph and in the whole third paragraph. Second paragraph was improved I terms of referencing. It is crucial for a review paper that references are clear and robust for each statement. I would also recommend the authors to mentioned in the text that the biofortified millet has not been released in other parts of the world, as stated in the cover letter.

With my best regards

Author Response

reviewer comments:

Abstract

The abstract is still too long. It has more than 400 words and sounds more like a short bibliography review. The points raised in the abstract are interesting and come in a good sequence. Although, I think it could be shorten to around 250 words using more direct sentences and linking the ideas better together. It would make the abstract to be easier to read and call the attention of the reader. There are some sentences with lack of sense and grammatical problems and the English must be reviewed.

Response; Abstract revised as per suggestions (247words).   

Introduction

There are still sentences without references in the introduction, mainly at the end of the first paragraph and in the whole third paragraph. Second paragraph was improved I terms of referencing. It is crucial for a review paper that references are clear and robust for each statement. I would also recommend the authors to mentioned in the text that the biofortified millet has not been released in other parts of the world, as stated in the cover letter.

Response; thanks for suggestion. Now we stated as first release which mean nowhere else on this crop.  
